# Detailed Courses and Pathological Findings of Colonic Perforation without Diverticula in Sisters with Musculocontractural Ehlers–Danlos Syndrome Caused by Pathogenic Variant in *CHST14* (mcEDS-*CHST14*)

**DOI:** 10.3390/genes14051079

**Published:** 2023-05-14

**Authors:** Tomoko Kobayashi, Fumiyoshi Fujishima, Kazuaki Tokodai, Chiaki Sato, Takashi Kamei, Noriko Miyake, Naomichi Matsumoto, Tomoki Kosho

**Affiliations:** 1Department of Pediatrics, Tohoku University Hospital, Sendai 980-8574, Japan; 2Tohoku Medical Megabank Organization, Tohoku University, Sendai 980-8573, Japan; 3Department of Pathology, Tohoku University Hospital, Sendai 980-8574, Japan; 4Department of Surgery, Tohoku University Hospital, Sendai 980-8574, Japan; 5Department of Human Genetics, Yokohama City University Graduate School of Medicine, Yokohama 236-0004, Japan; 6Department of Human Genetics, Research Institute National Center for Global Health and Medicine, Tokyo 162-8655, Japan; 7Department of Medical Genetics, Shinshu University School of Medicine, Matsumoto 390-8621, Japan; 8Center for Medical Genetics, Shinshu University Hospital, Matsumoto 390-8621, Japan; 9Division of Clinical Sequencing, Shinshu University School of Medicine, Matsumoto 390-8621, Japan; 10Research Center for Supports to Advanced Science, Shinshu University, Matsumoto 390-8621, Japan

**Keywords:** musculocontractural Ehlers–Danlos syndrome (mcEDS), carbohydrate sulfotransferase 14 (*CHST14*), gastrointestinal perforation, diverticulum

## Abstract

Musculocontractural Ehlers–Danlos syndrome (mcEDS) is a heritable connective tissue disorder characterized by multiple congenital malformations and progressive connective-tissue-fragility-related manifestations in the cutaneous, skeletal, cardiovascular, visceral, ocular, and gastrointestinal systems. It is caused by pathogenic variants in the carbohydrate sulfotransferase 14 gene (mcEDS-*CHST14*) or in the dermatan sulfate epimerase gene (mcEDS-*DSE*). As gastrointestinal complications of mcEDS-*CHST14*, diverticula in the colon, small intestine, or stomach have been reported, which may lead to gastrointestinal perforation, here, we describe sisters with mcEDS-*CHST14*, who developed colonic perforation with no evidence of diverticula and were successfully treated through surgery (a resection of perforation site and colostomy) and careful postoperative care. A pathological investigation did not show specific abnormalities of the colon at the perforation site. Patients with mcEDS-*CHST14* aged from the teens to the 30s should undergo not only abdominal X-ray photography but also abdominal computed tomography when they experience abdominal pain.

## 1. Introduction

Musculocontractural Ehlers–Danlos syndrome (mcEDS) is a rare subtype of EDS, caused by biallelic pathogenic variants in the carbohydrate sulfotransferase 14 gene (*CHST14)* encoding dermatan 4-*O*-sulfotransferase 1 (mcEDS-*CHST14*; MIM#601776) or in the dermatan sulfate epimerase gene (*DSE*) (mcEDS-*DSE*; MIM#615539) [1,2,3]. To date, 67 patients (49 families) with mcEDS-*CHST14* and 14 patients (eight families) with mcEDS-*DSE* have been reported [4,5,6,7,8]. Patients with mcEDS typically present with multiple congenital malformations (e.g., craniofacial characteristics, congenital multiple contractures, and congenital visceral/ocular abnormalities) and progressive multisystem-fragility-related manifestations (e.g., skin hyperextensibility/fragility/bruisability, joint hypermobility/dislocation, spinal/foot deformities, and large subcutaneous hematomas) [1,2,3]. In a recently reported large international collaborative series on mcEDS-*CHST14*, the identified gastrointestinal features included constipation (most commonly), diverticula, abdominal pain, diarrhea, large bowel sounds, and gastric ulcers [4]. Diverticula occurred in eight patients (35%), mostly in the colon or small intestine, four of whom developed perforation at ages ranging from 12 to 29 years [4].

Here, we describe detailed clinical and pathological findings of two sisters with mcEDS-*CHST14*, both of whom developed colonic perforation without any diverticula in their 20s. This report could have implications for the pathomechanism of colonic perforation as well as for the diagnostic and therapeutic strategy for this lesion in patients with mcEDS.

## 2. Materials and Methods

### 2.1. Case Presentation

Two sisters with mcEDS-*CHST14* were recruited in this study. Clinical and radiological information was obtained from their medical records at Tohoku University Hospital. Their past histories were included in a recently reported large international collaborative study on mcEDS-*CHST14*, in which no individual clinical information or history was described [4]. Written informed consent for participating in this study was obtained from the patients.

Patient 1, the older sister, is currently 29 years old. In early childhood, she underwent orthopedic surgical treatment for bilateral clubfeet and a left hip dislocation. During her schooldays, she developed recurrent dislocations in bilateral patellae and the right shoulder, requiring manual reduction. She also repeatedly developed large subcutaneous hematomas on her extremities from minor traumas, which recovered spontaneously after about 3 weeks. The clinical course of the disease was complicated by pyelonephritis at age 15 years, gastritis at age 19, and left ureteral stones at age 20, all of which were treated with pharmacological therapy. She had severe constipation, with defecation only once every 7 days, and her stools varied from hard to soft. After graduating from college, she worked in an office for a company. At age 16 years, she was suspected of having mcEDS clinically, and was found to have a recurrent variant in *CHST14* (NM_130468.3:c.842C>T:p.Pro281Leu) homozygously through Sanger sequencing.

Patient 2, the younger sister, is currently 27 years old. In childhood, she underwent correction of bilateral flat valgus feet with braces. Deformity and rigidity of the ankle joint persisted, and she sprained her left ankle three times, complicated by subcutaneous hematomas and skin laceration requiring surgical suture. She repeatedly developed large hematomas on her extremities from minor traumas, which recovered spontaneously after about 3 weeks. She developed five dislocations of the left shoulder and one dislocation of the right shoulder during her schooldays. At age 13 years, she underwent corrective surgery for progressive scoliosis. At age 17 years, she fell and dislocated her left hip, which was reduced manually. Bilateral retinal detachments occurred at age 18 years, which were treated with photocoagulation. She had moderate constipation, with defecation once every 3 to 4 days, and her stools varied from hard to soft. She suffered from acute gastritis at age 24 years and gastroesophageal reflux disease at age 26, both of which were treated pharmacologically. After graduating from university, she worked in an office. At age 14 years, she was suspected of having mcEDS clinically, and was found to have the same homozygous variant in *CHST14* as her sister through Sanger sequencing.

### 2.2. Pathological Investigation

Paraffin-embedded samples of a perforated tissue and a normal tissue adjacent to it in the transverse colon of Patient 2 were obtained. Hematoxylin–eosin staining, immunochemical staining for desmin, and elastica-Masson staining were performed in accordance with the standard protocols. Written informed consent for the publication of this case report was obtained from the patients.

## 3. Results

### 3.1. Detailed Course of Colonic Perforation

Patient 1 visited our emergency department due to abdominal pain after 2 weeks of constipation at age 25 years. After administering an enema, the abdominal pain progressed and her blood pressure decreased. Abdominal X-ray photography (Xp) failed to determine the cause of the abdominal pain, while contrast-enhanced computed tomography (CT) showed free air in the abdominal cavity and ascite accumulation (Figure 1A). Emergency surgery was performed based on the diagnosis of perforation of the digestive tract accompanied by panperitonitis. A large amount of ascites including stools and food residues was observed, indicating that the perforation had occurred in the lower gastrointestinal tract. The perforation site, slightly shorter than 3 cm in length, was found to be located slightly on the left side of the transverse colon. No obvious cause of the perforation could be identified. After washing the abdominal cavity with more than 10 L of saline, the perforated site was elevated and a colostomy was created in a double-barreled fashion (Figure 2A). After the surgery, she was managed in the intensive care unit and treated intensively with antibiotics, immunoglobulin, steroids, noradrenaline, and pitressin for septic shock, along with hemoperfusion with continuous hemodiafiltration and polymyxin-B-immobilized fiber. Although she was weaned from the ventilator on postoperative day (POD) 2, she developed respiratory failure on POD 3 and was placed on a ventilator again. A tracheostomy was performed on POD 6, and she was again weaned off the ventilator on POD 13. The tracheostomy was closed on POD 32 as her respiratory distress resolved. She was transferred to the rehabilitation department from the surgical department on POD 49. She underwent rehabilitation for disuse syndrome for 1 month. Specifically, she received training for the activities of daily living within a range that does not cause joint pain or fear of falling. Three months after the onset, she was discharged home and recovered enough to return to work with a colostomy.

Patient 2 had mild pain in her upper left abdomen from the morning at age 24 years. She had a bowel movement after eating lunch, which relieved her abdominal pain. Because her abdominal pain became worse in the evening, she left work early and visited our hospital. Abdominal Xp showed no abnormal findings, but the possibility of gastrointestinal perforation was considered based on her past history with acute gastritis as well as Patient 1’s episode. Contrast-enhanced CT showed a small amount of intra-abdominal free air (Figure 1B), so she was admitted to the surgical unit. An upper gastrointestinal endoscopy showed no abnormality in the upper gastrointestinal tract; therefore, perforation of the lower gastrointestinal tract was suspected, and emergency surgery was performed. The intra-abdominal contamination was relatively mild and a perforation was observed in the colon at the splenic flexure. After the perforation site was resected, the abdominal cavity was flushed with a large amount of saline solution, and a colostomy of the transverse and descending colon was created in a double-barreled fashion (Figure 2B). The postoperative course was complicated by a wound infection and an intra-abdominal abscess, which improved with antibiotics. She was discharged home and recovered enough to return to work with a colostomy on POD 23.

**Figure 1 genes-14-01079-f001:**
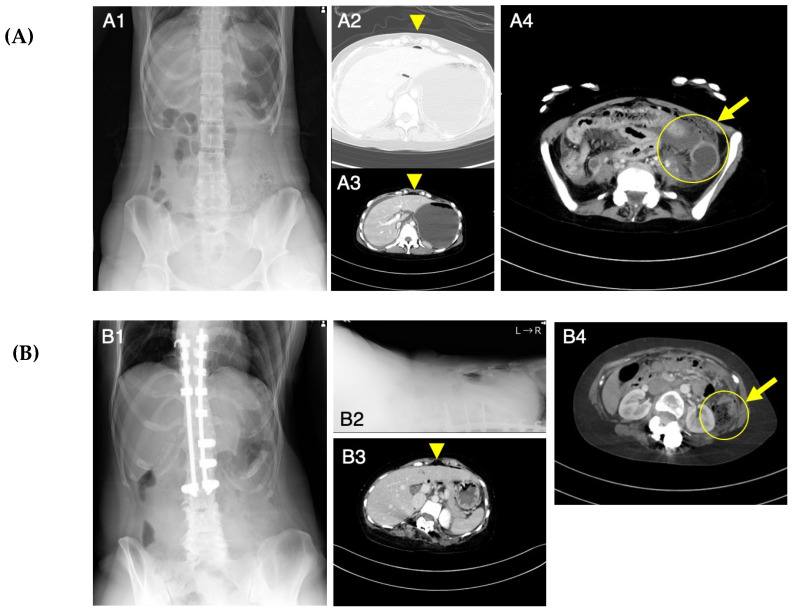
Abdominal X-ray photography (Xp) and computed tomography (CT) images at the onset of acute abdomen in Patients 1 (**A**) and 2 (**B**). Abdominal Xp in a standing position showed no abnormal findings (**A1**). Abdominal CT revealed intraperitoneal free air (**A2**, **A3**) and findings suggestive of a perforation site (**A4**). Abdominal Xp in standing (**B1**) and supine (**B2**) positions showed no abnormal findings. Abdominal CT revealed a small amount of free air on the surface of the liver (**B3**) and findings suggestive of a perforation site (**B4**). Arrowheads indicate free air and arrows indicate a possible perforation site.

**Figure 2 genes-14-01079-f002:**
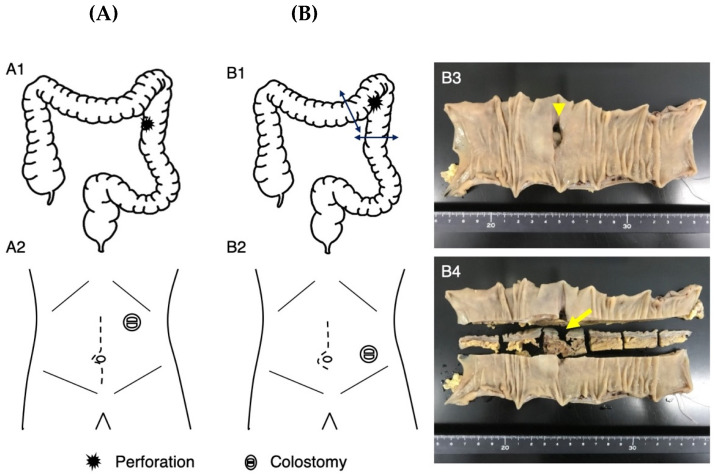
Surgical details of Patients 1 (**A**) and 2 (**B**). In Patient 1, the perforated site of the colon (**A1**) was elevated outside the left upper abdominal wall and a colostomy was constructed (**A2**). In Patient 2, the perforation site of the colon at the splenic flexure was resected using a linear stapler (**B1**). A colostomy of the transverse and descending colon was created at the left lower abdominal wall (**B2**). Macroscopic observation showed the perforation site ((**B3**), arrowhead) and its sectional view ((**B4**), arrow).

### 3.2. Pathological Investigation

Histopathologically (Figure 3), the muscularis propria was ruptured at the perforated area, but no microscopic morphological changes could be detected in the smooth muscle, and no obvious changes were observed in the surrounding vessels. There were no findings suggestive of the cause of the perforation. The layered structure of the background intestinal wall was also well-preserved, and no obvious diverticulum was observed.

## 4. Discussion

The sisters aged 25 and 24 years with mcEDS-*CHST14* as described in this paper developed spontaneous colonic perforation with no evidence of diverticula or of no apparent macroscopic or pathological abnormalities of the colon and were treated successfully with no sequelae. We cannot exclude the possibility of additional gene mutations that could contribute to the severity of findings in these women because they were diagnosed not through comprehensive genetic screening such as whole exome sequencing but through targeted sequencing.

Previously reported patients with mcEDS-*CHST14* who developed colonic perforation were complicated by diverticula [4], suggesting that diverticula confer susceptibility to perforation, although no clear pathological evidence of this has been identified [4,7,8]. A recently described patient, a 36-year-old male with mcEDS-*CHST14*, developed colonic perforation at the splenic flexure 11 days after corrective surgery for talipes equinovarus, and underwent an emergency laparotomy plus partial colectomy with proximal fistulotomy [5]. Recovery from subsequent septic shock was achieved through anti-infective therapy, nutritional support, blood transfusion, and mechanical ventilation. Ten days after the perforation, he developed a perforation of the gastric antrum, followed by septic shock. Eight days after the gastric perforation, he developed another possible intestinal perforation, complicated by a third septic shock, resulting in multiple organ failure and death [5]. Diverticula were not mentioned, and pathological study showed the infiltration of a large number of inflammatory cells and abscess formation on the first colonic perforation site and inflammatory cell infiltration with fibroblast proliferation in the fibrous connective tissue at the gastric perforation site [5]. It was speculated that his colonic perforation had been caused by tissue and muscle fragility, and worsened by weakened gastrointestinal peristalsis and secondary intestinal obstruction caused by long-term bed rest after orthopedic surgery [5]. The sisters we have described in this report are therefore the first to be demonstrated not to have diverticula relevant to their colonic perforations, but no pathological abnormalities causative of the perforations were identified.

To the best of our knowledge, no systematic investigation has been performed on histopathological abnormalities of the gastrointestinal system in patients with mcEDS. Microscopic and ultrastructural abnormalities of the skin have been systematically investigated: light microscopy—fine collagen fibers predominantly present in the reticular to papillary dermis and a marked reduction of normally thick collagen bundles; transmission electron microscopy—dispersed collagen fibrils in the papillary to reticular dermis (in contrast to healthy controls showing regularly and tightly assembled collagen fibrils); and transmission electron microscopy with cupromeronic blue staining to visualize glycosaminoglycan (GAG) side chains of decorin—linear, stretched GAG chains from the outer surface of collagen fibrils to adjacent fibrils (in contrast to healthy controls showing curved GAG chains maintaining close contact with attached collagen fibrils) [9]. The skin features (marked hyperextensibility and fragility) are thought to be caused by a disorganized collagen fibril network through these structural changes of decorin GAG side chains resulting from compositional changes of GAG side chains from dermatan sulfate to chondroitin sulfate due to defects of dermatan 4-*O*-sulfotransferase 1 or dermatan sulfate epimerase [9,10]. Because the gastrointestinal features tend to be milder than the skin features [4], it might be difficult to demonstrate the microscopic abnormalities of the gastrointestinal system. However, transmission electron microscopy with cupromeronic blue staining might be able to demonstrate ultrastructural abnormalities of the collagen fibril network in the gastrointestinal system, although no such samples were available in the sisters reported here.

Including the data for the sisters reported here, the age of onset of gastrointestinal perforation in mcEDS tends to range from the teens to the 30s [4]. Abdominal Xp failed to detect gastrointestinal perforation when the sisters reported here experienced abdominal pain. Therefore, patients with mcEDS aged from the teens to the 30s should undergo abdominal CT upon suspicion of gastrointestinal perforation.

Further reports on gastrointestinal perforation in patients with mcEDS are needed to uncover the pathomechanisms of this critical complication as well as to establish the management strategy, including prevention (e.g., usefulness of treatment for constipation) and treatment (e.g., surgery and medication).

## 5. Conclusions

We have presented two sisters aged 25 and 24 years with mcEDS-*CHST14*, who developed spontaneous colonic perforation with no evidence of diverticula or of no apparent macroscopic or pathological abnormalities of the colon and who were treated successfully with no sequelae. The threshold for abdominal computed tomography in patients with mcEDS-*CHST14* aged from the teens to the 30s presenting with otherwise unexplained acute abdominal pain should be low so that spontaneous perforation is diagnosed early and immediately followed by antibiotic therapy and laparotomy.

## Figures and Tables

**Figure 3 genes-14-01079-f003:**
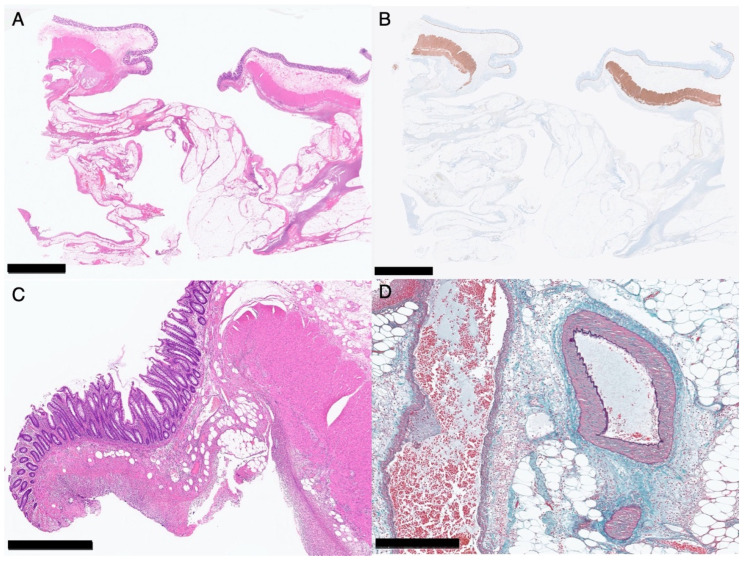
Histopathology of the perforated area of the colon of Patient 2. Loupe images ((**A**), hematoxylin–eosin staining; (**B**), immunological staining for desmin). Black bar indicates 5 mm. A medium-magnification image with hematoxylin–eosin staining (**C**). No morphological abnormalities are detected in the muscle fibers. Black bar indicates 1 mm. A high-magnification image with elastica-Masson staining (**D**). No abnormalities are detected in the vessel wall near the perforated area. Black bar indicates 500 μm.

## Data Availability

The data are available upon reasonable request.

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
