# Peer review of "Detailed Courses and Pathological Findings of Colonic Perforation without Diverticula in Sisters with Musculocontractural Ehlers–Danlos Syndrome Caused by Pathogenic Variant in CHST14 (mcEDS-CHST14)"

_genes, 2023, doi:10.3390/genes14051079_

Round 1

Reviewer 1 Report

This well-written paper is important for clinicians and expands the phenotypical spectrum of Musculocontractural Ehlers-Danlos Syndrome caused by pathogenic variants in CHST14 (mcEDS-CHST14).

I only have a few minor comments:

Line 73 : Please replace "She" with The clinical course of the disease...

Line 96: replace are with tissue

The authors might wish to add in the conclusions that ...the threshold for abdominal CT scan in patients presenting with otherweise unexplained acute abdominal pain should be low so that spontaneous perforation is diagnosed early and immediately followed by antibiotic therapy and laparatomy....

Author Response

Dear Reviewer 1

This well-written paper is important for clinicians and expands the phenotypical spectrum of Musculocontractural Ehlers-Danlos Syndrome caused by pathogenic variants in CHST14 (mcEDS-CHST14).

Response: Thank you so much for this high evaluation.

I only have a few minor comments:

Line 73 : Please replace "She" with The clinical course of the disease...

Response: Thank you. We have made the replacement, according to this comment.

Line 96: replace are with tissue

Response: Thank you. We have made the replacement, according to this comment.

The authors might wish to add in the conclusions that ...the threshold for abdominal CT scan in patients presenting with otherweise unexplained acute abdominal pain should be low so that spontaneous perforation is diagnosed early and immediately followed by antibiotic therapy and laparatomy....

Response: Thank you. We have added this comment to the last sentence of the conclusion.

Reviewer 2 Report

This detailed report on sisters with colonic perforation and DNA evidence of musculocontractural EDS presents important clinical information with imaging and histology studies that will add to the literature on EDS and bowel issues generally. Several suggestions should be considered by the authors.

1. I did not see the complete family history. While the homozygous mutations might imply no affected relatives, there are many cases of heterozygous mutations causing milder phenotypes of EDS, necessitating at least descriptions of parental findings.

2. The authors should mention the possibility of additional gene mutations contributing to the severity of findings in these women since targeted rather than whole exome sequencing seems to have been performed.

3. The GI complications are very typical of the low bowel motility of irritable bowel syndrome, itself a common finding in EDS due to adrenergic stimulation and cholinergic suppression as the autonomic system compensates for tissue laxity/lower body blood pooling and decreased cerebral perfusion. It would be useful to know if the patients had any signs of the food-medication intolerances of mast cell stimulation or the cardiac issues of POTS that can accompany this dysautonomia, perhaps playing a role in the severe hospital complications in their first patient.

Author Response

Dear Reviewer 2

This detailed report on sisters with colonic perforation and DNA evidence of musculocontractural EDS presents important clinical information with imaging and histology studies that will add to the literature on EDS and bowel issues generally. Several suggestions should be considered by the authors.

Response: Thank you so much for this high evaluation.

Point 1: I did not see the complete family history. While the homozygous mutations might imply no affected relatives, there are many cases of heterozygous mutations causing milder phenotypes of EDS, necessitating at least descriptions of parental findings.

Response 1: Parents with a heterozygous variant in CHST14 did not show relevant symptoms. The sisters have no other siblings.

Point 2: The authors should mention the possibility of additional gene mutations contributing to the severity of findings in these women since targeted rather than whole exome sequencing seems to have been performed.

Response 2: As Reviewer 2 commented, we cannot exclude the possibility of additional gene mutations contributing to the severity of findings in these women. We have added a sentence, “We cannot exclude the possibility of additional gene mutations that could contribute to the severity of findings in these women because they were diagnosed not through comprehensive genetic screening such as whole exome sequencing but through targeted sequencing.” in the discussion.

Point 3: The GI complications are very typical of the low bowel motility of irritable bowel syndrome, itself a common finding in EDS due to adrenergic stimulation and cholinergic suppression as the autonomic system compensates for tissue laxity/lower body blood pooling and decreased cerebral perfusion. It would be useful to know if the patients had any signs of the food-medication intolerances of mast cell stimulation or the cardiac issues of POTS that can accompany this dysautonomia, perhaps playing a role in the severe hospital complications in their first patient.

Response 3: Thank you for this valuable suggestion. None of the women had relevant signs of the food-medication intolerances of mast cell stimulation or the cardiac issues of POTS that can accompany this dysautonomia.
